# Measurement report: Vehicle-based Multi-lidar Observational Study of the Effect of Meteorological Elements on the Three-dimensional Distribution of Particles in the Western Guangdong–Hong Kong–Macao Greater Bay Area

Xinqi Xu[1,2], Jielan Xie[1,2], Yuman Li[1,2], Shengjie Miao[1,2], and Shaojia Fan[1,2]

[1]School of Atmospheric Sciences, Sun Yat-sen University, Zhuhai, 519082, China

[2]Guangdong Provincial Observation and Research Station for Climate Environment and Air Quality Change in the Pearl River Estuary, Key Laboratory of Tropical Atmosphere-Ocean System, Ministry of Education, Southern Marine Science and Engineering Guangdong Laboratory (Zhuhai), Zhuhai, 519082, China

*Correspondence to*: Shaojia Fan (eesfsj@mail.sysu.edu.cn)

**Abstract:** The distribution of meteorological elements has always been an important factor in determining the horizontal and vertical distribution of particles in the atmosphere. To study the effect of meteorological elements on the three-dimensional distribution structure of particles, mobile vehicle lidar, and fixed-location observations were collected in the western Guangdong–Hong Kong–Macao Greater Bay Area of China during September and October in 2019 and 2020. Vertical aerosol extinction coefficient, depolarization ratio, and wind and temperature profiles were measured using a micro pulse lidar, a Raman scattering lidar, and a Doppler wind profile lidar installed on a mobile monitoring vehicle. The mechanism of how wind and temperature in the boundary layer affects the horizontal and vertical distribution of particles was analysed. The results shows that particles were mostly distributed in downstream areas on days with moderate wind speed in the boundary layer, whereas they were distributed homogeneously on days with weaker wind. There are three typical types of vertical distribution of particles in the western Guangdong–Hong Kong–Macao Greater Bay Area (GBA): surface single layer, elevated single layer, and double layer. Analysis of wind profiles and Hybrid Single-Particle Lagrangian Integrated Trajectory Model(HYSPLIT)backward trajectory reveals different sources of particles for the three types. Particles concentrating near the temperature inversion and multiple inversions could cause more than one peak in the extinction coefficient profile. There were two mechanisms affecting the distribution of particulate matter in the upper and lower boundary layers. Based on this observational study, a general model of meteorological elements affecting the vertical distribution of urban particulate matter is proposed.

## 1. Introduction

The Guangdong–Hong Kong–Macao Greater Bay Area (GBA) is one of China's national key economic development regions. It consists of Guangzhou (GZ), Shenzhen (SZ), Zhuhai (ZH), Foshan (FS), Huizhou (HZ), Dongguan (DG), Zhongshan (ZS), Jiangmen (JM), and Zhaoqing (ZQ) in Guangdong province, as well as Hong Kong and Macao, the two Special Administrative Regions.

Covering 56,000 square kilometres, the GBA had a vast population of over 70 million at the end of
2018. The GBA plays a significant role in boosting global trade along the land-based Silk Road
Economic Belt and the 21st Century Maritime Silk Road. With the rapid development of the regional
economy, increasingly more studies on air quality and climate effect in the GBA have also been
conducted (Fang et al., 2018; Shao et al., 2020; Zhou et al., 2018).

Anthropogenic particles in the air play an important role in the environment of human living. They not
only act as air pollutants posing harmful effects to human health (Liao et al., 2017; Leikauf et al., 2020;
Yao et al., 2020; Orru et al., 2017) but also alter the temperature near the ground owing to their ability
to absorb and scatter solar radiation (IPCC, 2014; Strawa et al., 2010). As a result of industrialisation
and urbanisation, megacity clusters in China such as the Beijing–Tianjin–Hebei [also called Jing-Jin-Ji
(JJJ) in Chinese] area, Yangtze River Delta (YRD), and Guangdong–Hong Kong–Macao GBA, have
been seriously affected by particulate matter in recent years. Numerous studies on the particulate matter
have been conducted in these areas (Xu et al., 2018; Liu et al., 2017; Du et al., 2017). Particles in the
boundary layer can, directly and indirectly, affect human lives and activities. Therefore, it is essential
to study their distribution characteristics.

The distribution of particles is influenced not only by changes in source emissions but also by changes
in meteorological factors such as temperature and wind. It has previously been observed that a low
boundary layer height and complex vertical distributions of aerosols, temperature, and relative
humidity are the main structural characteristics of haze days (Huige et al., 2021). Previous studies have
confirmed that different types of temperature inversions have different impacts on particles in the
boundary layer (Wallace et al., 2009; Wang et al., 2018). The depth and temperature difference of the
inversion region is a key factor for predictions of surface $PM_{2.5}$ concentrations (Zang et al., 2017). It
has been previously observed that wind fields play an important role in transboundary-local aerosol
interactions (Huang et al., 2021a; Huang et al., 2021b). Recent evidence suggests that wind shear is an
important factor in terms of $PM_{10}$ vertical profile modification (Sekuła et al, 2021). The concentration
of particulate matter also shows characteristics of wind-dependent spatial distributions in which
pollutant transport within the GBA city cluster is significant (Xie et al., 2019). Hence, the issue of how
meteorological factors affect the distribution of particles has received considerable critical attention.

Lidar is an active remote sensing device. It emits a laser light beam and receives a backscatter signal,
which can be further used to retrieve the vertical distribution of particle optical properties, wind, and
temperature. It has been widely applied in the fields of meteorology and environmental science. In
most studies, it is used as a ground-based or satellite-based instrument (Tian et al., 2016; Liu et al.,
2017; Heese et al., 2017).

In recent years, vehicle-based lidar observation has been gradually developed and become a powerful
tool to detect the physical and chemical properties of the boundary layer. Compared with traditional
observations, it can carry out continuous mobile observations and obtain the change of vertical profiles
of certain factors in its path. Additionally, a mobile lidar system can be used to conduct supplementary
observations in areas with no lidar present. In the past few years, several vehicle-based observational
experiments have been carried out (Lv et al., 2017; Lyu et al., 2018; Lv et al., 2020; Zhao et al., 2021;
Fan et al., 2018), but research aimed at multi-lidar observations and the effect of the vertical structure
of meteorological factors on the distribution of particles has largely been an underexplored domain,
especially in the GBA. Former research revealed that pollution of particulate matter frequently occurs
in the western part of inland regions of GBA (Fang et al., 2019), affecting downstream cities under the
northerly wind field. Hence, the authors were motivated to perform observations in the western GBA
with a multi-lidar system installed on a vehicle to study the influence of the three-dimensional structure
of meteorological elements on the distribution of particles.
## 2. Data and Method
### 2.1 Description of Observations
The horizontal distribution of the particles was studied by making mobile vehicle lidar observations
over the west bank of the Pearl River Estuary. During the mobile vehicle lidar observations experiment,
the vehicle was driven clockwise along the west bank of the Pearl River Estuary, passing through main
cities of the GBA in the route, from as far north as Guangzhou to as far south as Zhuhai. The total
length of the route was approximately 320 km, and the experiment was conducted during the daytime.
The vehicle-based observation lasted for seven continuous days, which started on August 29th and
ended on September 4th, 2020. During most of the mobile observations, the relative humidity of
Zhuhai, the closest city to the sea, was below 60 %. Therefore, the influence of hygroscopic growth on
the extinction coefficient was negligible. To study the vertical distribution of the particles, we
conducted fixed-location lidar observation experiments using the same lidar system from September
10th to October 8th, 2019, and from August 29th to October 27th, 2020, totalling 89 days. The reason
for choosing these periods is that they include the wet season change to the dry season in the GBA area.
Therefore, changes in meteorological elements have a significant impact on the three-dimensional
distribution of particles. The location of the Haizhu Lake Research Base and the area of the measuring
path are shown in Fig. 1. The research area is on the Pearl River Delta Plain. This area is bordered by
the Nanling Mountains in the north. Mountain obstruction makes the GBA area less susceptible to
long-distance transport of pollutants from other areas, and the transport of pollutants mainly occurs
between cities in the research area. Observations with the vehicle-based multi-lidar system are listed in
Table 1.

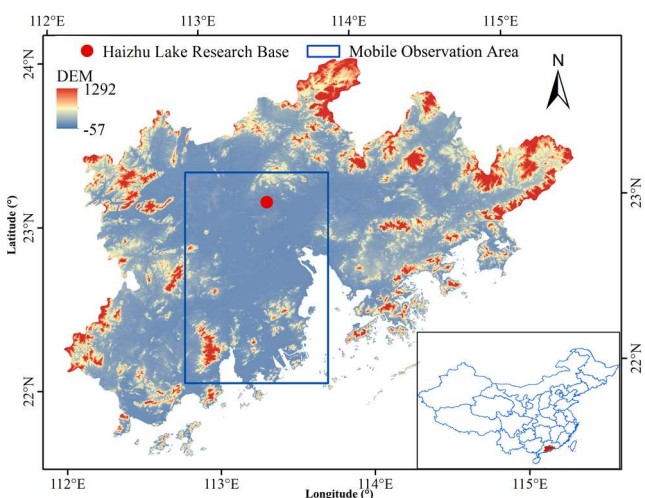

**Figure 1. Location of the Haizhu Lake Research Base and the mobile observation area.**

**Table 1. Observations with the vehicle-based multi-lidar system**

| Time | Observation |
|---|---|
| Sept. 10th – Oct. 8th, 2019 | Fixed-location observation |
| Aug. 29th – Sept. 4th, 2020, in the daytime | Mobile observation |
| Aug. 29th – Sept. 4th, 2020, at night | Fixed-location observation |
| Sept. 5th – Oct. 27th, 2020 | Fixed-location observation |


**2.2 Multi-lidar System**
A multi-lidar system was installed on a vehicle in this experiment. The car used was a modified 7-
seater Mercedes-Benz sport utility vehicle. Three lidars were fixed to the rear of the car by steel bars to
ensure their stability. To avoid the impact of frequent changes in speed and vehicle bumps during the
observation, the routes of mobile observations were basically flat highways, and the driving speed was
controlled within 80 km/h. During fixed-location observations, the car was parked in the observation
field and connected to a stable power source. The lidar system included a 3D visual scanning micro
pulse lidar (EV-Lidar-CAM, EVERISE Company, Beijing,
http://www.everisetech.com.cn/products/ygtc/evlidarportable.html), a twirling Raman temperature
profile lidar (TRL20, EVERISE Company, Beijing,
http://www.everisetech.com.cn/products/ygtc/templidar.html), a Doppler wind profile lidar
(Windview10, EVERISE Company, Beijing,
http://www.everisetech.com.cn/products/ygtc/windview10.html), a global positioning system (GPS),
and a signal acquisition unit. The three lidars are characterised by high temporal and spatial resolution
and can effectively determine the evolution of the vertical distribution of particles, as well as
temperature, wind speed, and wind direction over time. The quality of data from the lidar system was
checked before using in our study. Results show that the percentage difference between data provided
by the lidar system and data from the Shenzhen meteorological tower was less than 15%, which
indicates a sufficient accuracy of the lidar instrument. We have used this lidar system in our previous
research and showed it to be reliable (He et al., 2021a; He et al., 2021b). The vehicle setup is shown in
Figure 2. Details of the three lidars are shown in Table 2.

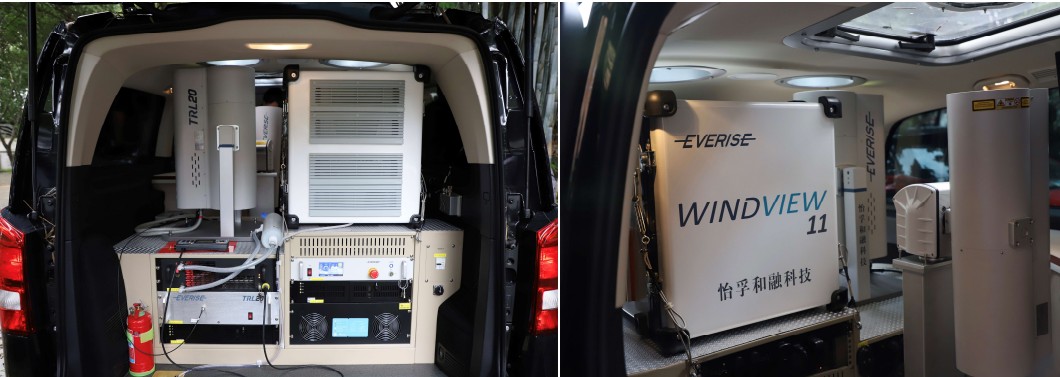
**Figure 2. Setup of the multi-lidar system on the vehicle.**

**Table 2. Detailed parameters for the three lidars.**

| Lidar | Variable | Laser source | Wave length | Laser frequency | Spatial resolution | Time resolution |
|---|---|---|---|---|---|---|
| Micro pulse lidar | Original signal, Extinction coefficient profiles, Depolarization ratio profiles, Aerosol optical depth | Nd:YAG laser | 532 nm | 2500 Hz | 15 m | 1 min |
| Raman temperature profile lidar | Temperature profiles | Nd:YAG laser | 532 nm | 20 Hz | 60 m | 5 min |
| Doppler wind profile lidar | Wind speed profiles, Wind direction profiles | Fibre pulse laser | 1545 nm | 10 kHz | 50 m | 1 min |


## 2.3 Calculation of Extinction Coefficient and Depolarization Ratio

The aerosol extinction coefficient represents the reduction of radiation in a band owing to scattering
and absorption by aerosols (Li et al., 2020). The formula for the extinction coefficient calculation
(Fernald, 1984) is as follows:

$$\alpha_a(z) = -\frac{S_a}{S_m}\alpha_m(z) + \frac{P(z)z^2 \cdot \exp\left[2\left(\frac{S_a}{S_m}-1\right)\int_z^{z_c}\alpha_m(z)\mathrm{d}z\right]}{\frac{P(z_c)z^2}{\alpha_a(z_c)+\frac{S_a}{S_m}\alpha_m(z_c)}+2\int_z^{z_c}P(z)z^2\exp\left[2\left(\frac{S_a}{S_m}-1\right)\int_z^{z_c}\alpha_m(z)\mathrm{d}z\right]\mathrm{d}z} \qquad (1)$$

where $P(z)$ is the power received at altitude $z$, $\alpha_a$ and $\alpha_m$ the particle extinction and molecular
extinction, respectively and $S_a = 50\,\mathrm{Sr}$ the particle extinction-to-backscatter ratio, which is the
default value given by the manufacturer. This value is consistent with prior work in the GBA area (Li et
al., 2020). $S_m = 8\pi/3$ is the molecular extinction-to-backscatter ratio, and $z_c$ the calibration height
of the micro pulse lidar, which is variable, ranging from 10-15 km, and depending on the signal
intensity.

The micro pulse lidar (MPL) system uses the scattering of polarized light to distinguish between
spherical and non-spherical particles to ascertain the particle species (Li et al., 2020). The
depolarization ratio is calculated with the following formula:

$$\delta = k\frac{P_\perp}{P_\parallel} \qquad (2)$$

where $P_\perp$ and $P_\parallel$ represent the cross-polarized and co-polarized signal, respectively. k the
depolarization calibration constant, which is the ratio of the gains of the parallel and perpendicular
channels (Dai et al., 2018).

## 2.4 HYSPLIT Backward Trajectory Model

The regional transport of particulate matter was studied using the National Oceanic and Atmospheric Administration Hybrid Single-Particle Lagrangian Integrated Trajectory Model (HYSPLIT) so as to determine the trajectory of air masses. It has been widely used in the field of air masses and pollutant source analysis (Deng et al., 2016; Lu et al., 2018; Kim et al., 2020). In this study, meteorological data of the Global Data Assimilation System (GDAS) at the spatial resolution of 0.25° was used. To obtain the sources of particulate matter at different heights, altitudes of 100 m, 500 m, and 1000 m were set as the ending points of the trajectories.

## 3. Results and Discussion

## 3.1 Mobile Vehicle Lidar Observations

The horizontal distribution of particles was obtained by conducting mobile vehicle lidar observations in the GBA. The reason for choosing this route is that it covers the major urban agglomerations in the western part of the Guangdong–Hong Kong–Macao Greater Bay Area, which contains a large number of anthropogenic aerosol emission sources. It is representative of the regional distribution of particles in this area. We conducted mobile observations once a day, from August 29th to September 4th, 2020. The set off time was at 10:00 and a single measurement circle was completed at around 16:00. Owing to surface heating, convection in the boundary layer develops vigorously during daytime, which allows aerosols to mix well and form a more homogeneous vertical distribution. Therefore, mobile observations during the daytime are more appropriate to study the horizontal distribution of particles in the GBA area. Figure 3 shows the aerosol optical depth (AOD) measured with the MPL in the route. Because of GPS signal interference, some GPS data on August 31st and September 2nd were missing. On most days, sections with high AOD values fell geographically into the south and west sides of the observation region. Figure 4 shows low-level horizontal wind fields on 925 hPa over the region based on ERA5 reanalysis data. In the first three days, the wind speed over the GBA was generally higher, with an easterly and north easterly direction. Polluted aerosols were transported along with the wind to the west and south of the study area. They accumulated in the downstream area, resulting in a high value of AOD. On September 1st, 3rd, and 4th, the GBA was in an area of low wind speed, which was not conducive to the regional transport of particulate matter. As a result, the AOD value of the whole GBA reached a higher level, of which the increase in AOD in the northern region was more obvious. AOD values on these days distributed more homogenously than days with higher wind speed. On September 2nd, the lower winds of the GBA turned westerly when the observation area in the east was downstream, and the highest points of the AOD value also appeared on the eastern route. Such results show that the horizontal distribution of particles in the GBA was closely related to wind speed and wind direction.

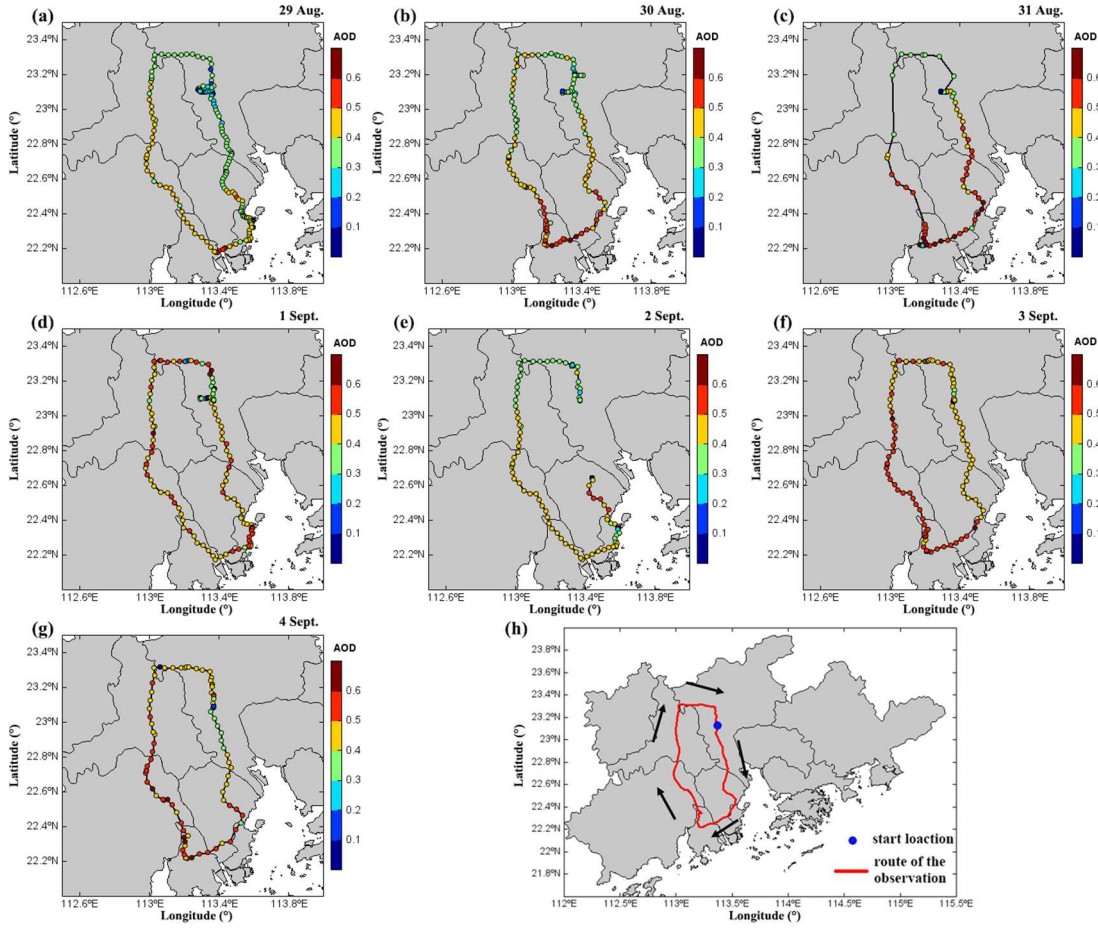



**Figure 3. (a)-(g) Aerosol optical depth (AOD) measured with the MPL in the route from August 29th to**
**September 4th, 2020, and (h) Guangdong–Hong Kong–Macao Greater Bay Area and route details.**

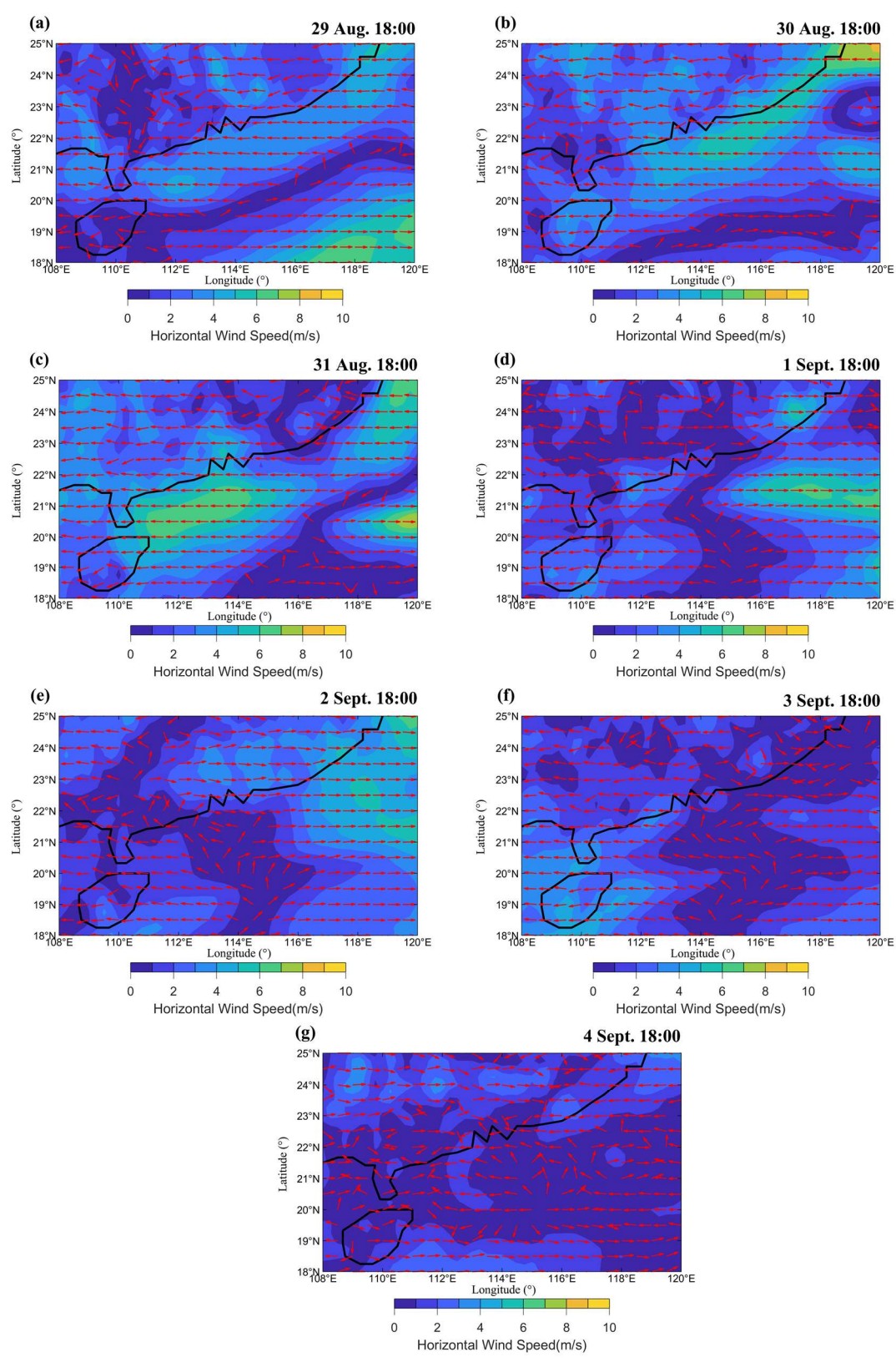



**Figure 4. (a)-(g) Wind field of 925 hPa from August 29th to September 4th, 2020. The colour map represents**
**horizontal wind speed (m/s). Red arrows represent the wind direction.**

### 3.2 Fixed-location Lidar Observations

To obtain the vertical distribution of particles, fixed-location lidar observations were conducted at the Haizhu Lake Research Base, which is located in the centre of the metropolis Guangzhou. The research base is representative of the distribution of urban aerosols. Unfortunately, there is no remote sensing device in the base. This motivated us to park the car in the base and conduct a total of 89 days of fixed-location observation. During this period, we found that the hierarchical structure of aerosols occurred more frequently at night, and most of the vertical aerosol distributions are consistent with three distribution types. Therefore, we selected the three most representative processes for analysing the three distribution types. Three different vertical distribution types of particles are given below, as well as the corresponding vertical observation results of temperature and wind in the same period. Altitude values in the following figures refer to the altitude above instrument.

#### 3.2.1 Type I: Surface Single Layer

On September 3rd, 2020, a clear night in autumn, the lidar system operated from 2154 to 0609 local time (LT) the next day. Figure 5(a) shows the time series of the extinction coefficient of a single aerosol layer on the surface, which was observed with the MPL. Before 0300 LT, particles accumulated below 800 m. The maximum value of the extinction coefficient near the ground was between 0.3–0.5. During 0300 LT and 0400 LT, there is a significant increase in the maximum height of the particle layer. After 0430 LT, the maximum height of the particle layer dropped, and the near-ground extinction coefficient fell below 0.3. Figure 5(b) shows the time series of corresponding depolarization ratio profiles. Most of the depolarization ratios were below 0.1, consistent with previous research on the GBA (Tian et al., 2017). A layer of elevated depolarization ratio was visible near the boundary of the surface single layer in Figure 5(b). It can be seen that during 0300 LT and 0400 LT, there was a significant hierarchical structure with a high depolarization ratio layer near the ground and another layer of high value above. A layer with a lower value of depolarization ratio existed between the two layers with a higher value. This result indicated that there might be local anthropogenic emissions during the period.

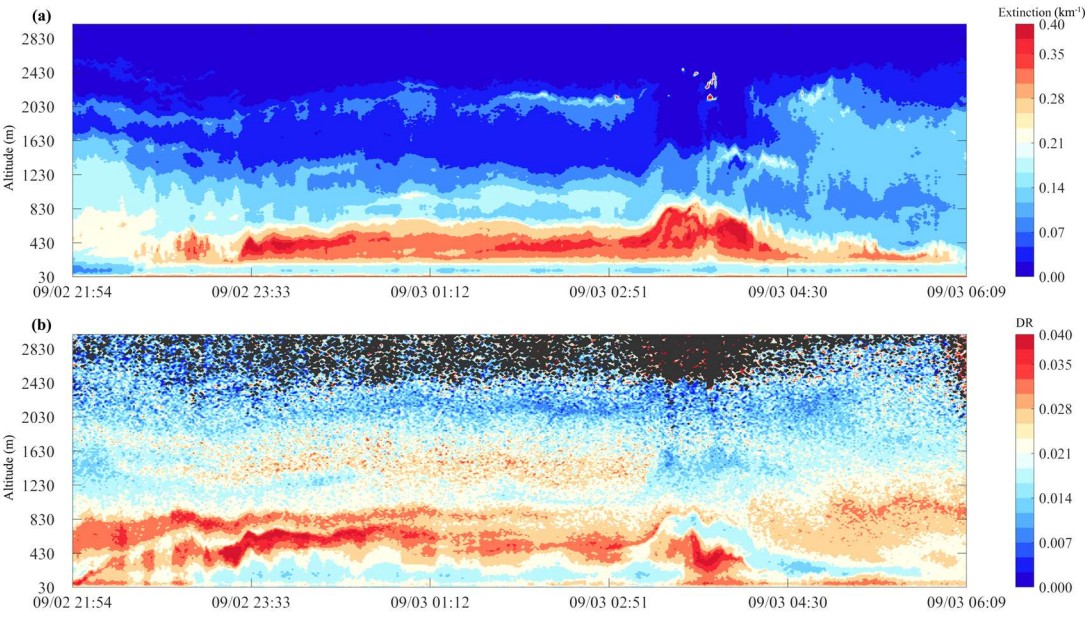

**Figure 5. Extinction coefficient at 532 nm (a) and depolarization ratio (b) from 2154 LT on September 2nd, 2020, to 0609 LT on September 3rd, 2020.**


Figure 6 shows the horizontal wind speed and wind direction in this period. Noticeably, a light wind
layer appeared below 1000 m, with horizontal wind speeds of each height maintained below 2 m/s.
Such a static and stable condition was advantageous to the accumulation of locally generated
particulate matter near the ground. However, light wind at higher altitude (500–1000 m) prevented the
regional transport of particulate matter at a higher altitude, because it is difficult for such a low wind
speed to blow the particulate matter at the corresponding height to the downstream area. Therefore,
when light wind dominated near the ground, the particulate matter was likely to form a single layer on
the surface.

It is worth noting that the wind at an altitude of 540 m at night gradually shifted to southerly wind,
whereas the northerly weight of the 290 m altitude wind gradually increased. This shift in the wind was
typical of a sea-land breeze in nocturnal coastal areas, which can only be observed when the
background wind speed was relatively low.

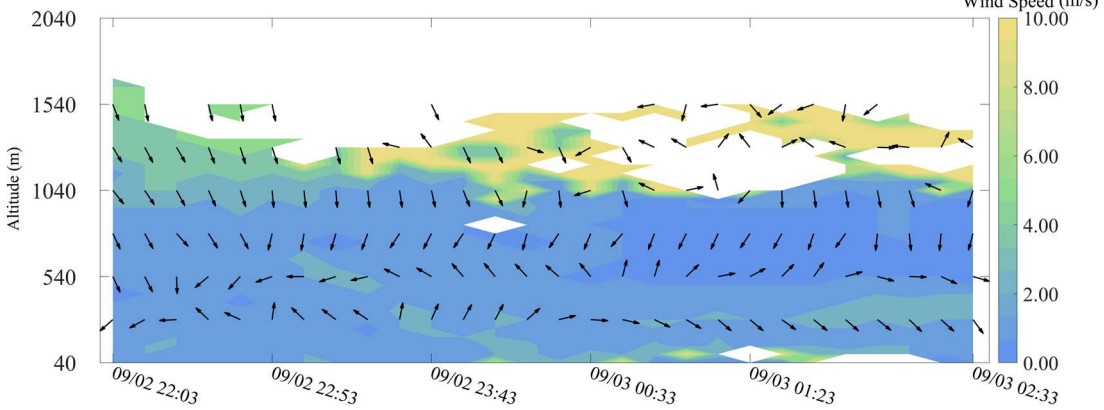


**Figure 6. Wind speed and wind direction of Type I. Colour map represents horizontal wind speed (m/s).**
**Arrows represents the wind direction.**

The backward trajectories analysis of the same period (Figure 7) shows that on a large scale, the
airflow in the boundary layer came from the north. The vertical trajectories of each layer were roughly
parallel within 24 h, and all moved from high altitude to low, suggesting that particulate matter emitted
near the ground in neighbouring cities was not easily transported by wind to Guangzhou.

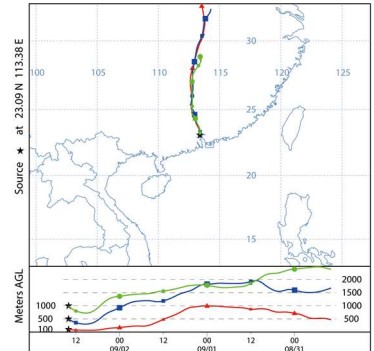


**Figure 7. Backward trajectories at 100 m, 500 m, and 1000 m, ending at 2200 LT September 2nd, 2020,**
**determined by the HYSPLIT model.**

Observations from the Raman temperature profile lidar (Figure 8) show an inversion between 600–
1200 m before 0300 LT, which then rose to 1200 m and shrank to near the ground. Temperature
inversion often exists at the top of the planetary boundary layer, trapping moisture and aerosols (Seibert
et al., 2000). Hence, changes in the height of the inversion coincided with the trend of the top of the
particulate matter layer on the vertical dimension revealed by MPL.

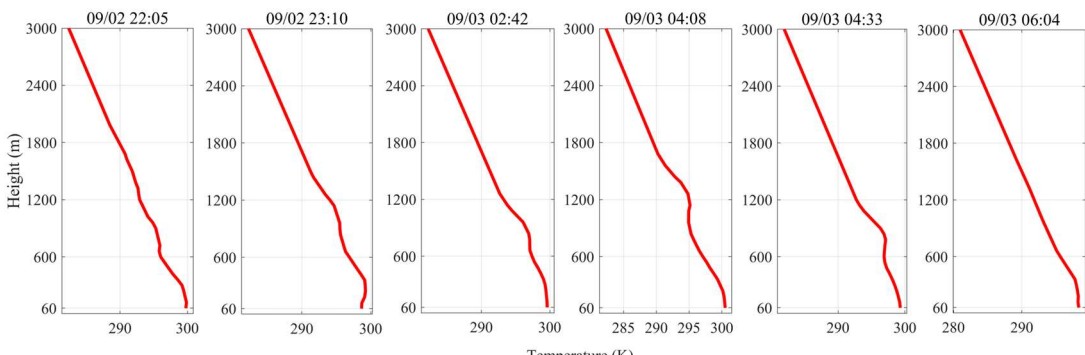


**Figure 8. Temperature profiles from the evening of September 2nd, 2020 to the early hours of September 3rd,**
**2020.**
**3.2.2 Type II: Elevated Single Layer**
The particle layer was not only distributed near the ground but sometimes suspended at a higher
altitude. Figure 9(a) shows the extinction coefficient time series of an elevated single layer of
particulate matter. The low extinction coefficient near the ground suggests that it was clean below 400
m during the night. The height of the high extinction coefficient layer gradually rose from 500–800 m
at night, which then dropped below 400 m after dawn. The high value of the extinction coefficient
corresponded to a higher depolarization ratio than the lower layer, which was approximately 0.02.
However, the depolarization ratio of *Type II* was significantly lower than the depolarization ratio of the
particle layer near the surface of *Type I*. This differing depolarization ratio was because local emissions
dominated in *Type I*, and the primary pollutant emissions from anthropogenic sources near the surface
with a non-spherical character and larger particle size accounted for a larger amount than that of *Type*
*II*.

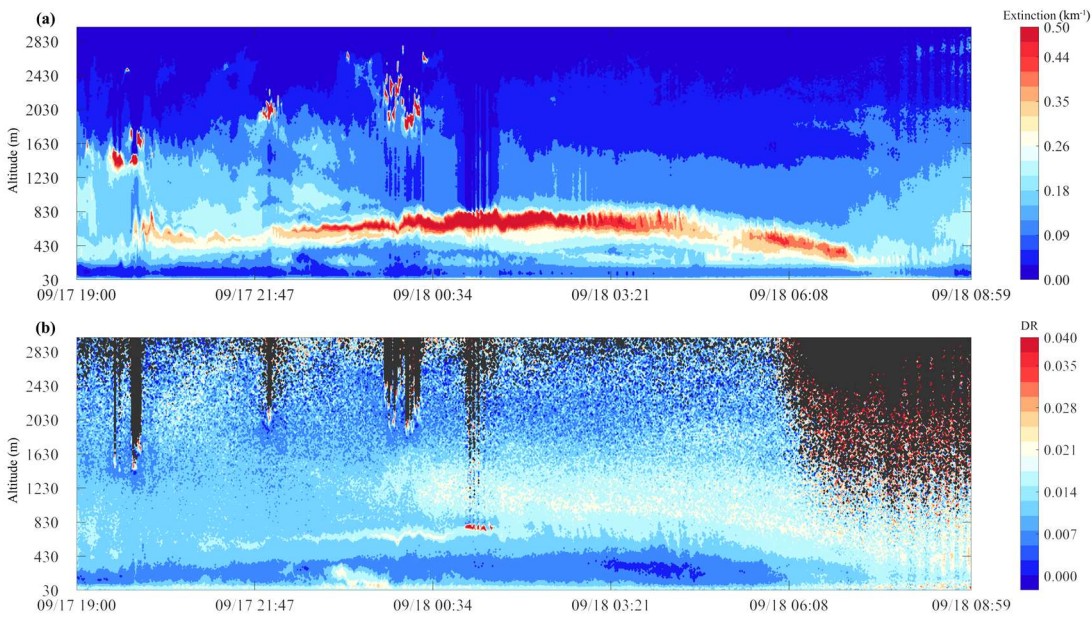

**Figure 9. Extinction coefficient at 532 nm (a) and depolarization ratio (b) from 1900 LT on September 17th, 2019 to 0859 LT on September 18th, 2019.**

Figure 10(a) indicated that backward trajectories at 500 m and 100 m were both from near the ground, elevating particles from lower levels vertically. Meanwhile, lower trajectories also carried particles from the upper reaches of the region over Guangzhou horizontally. The domination of weak wind in the boundary layer was beneficial to inter-city transport of particles. It brought particles from cities located upstream to the location of our observation and allowed particles to stay longer without being blown quickly downstream. In contrast, the trajectory at 1000 m came from a distance in the Y R D with a larger wind speed, and the trajectory remained at a high altitude. Particles at 1000 m cannot stay for a long time and were quickly transported downstream by strong winds. Hence, upward airflow near the ground and vertical wind shear at a higher altitude were the causes of particulate matter forming an elevated single layer. Unfortunately, the temperature profile and wind profile data were missing owing to sampling failures. This upward convection of particles was confirmed by the ERA5 vertical velocity reanalysis data of the corresponding time, shown in Figure 11.

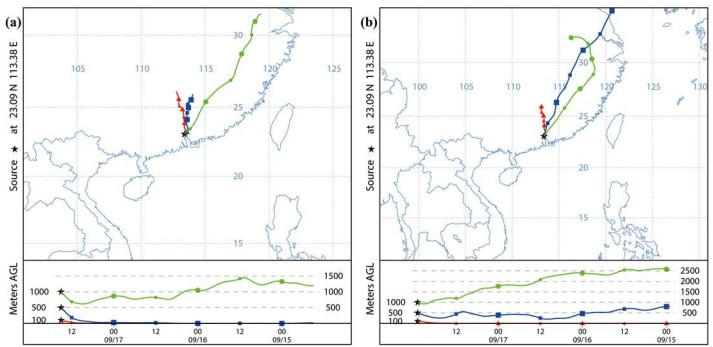

**Figure 10. Backward trajectories at 100 m, 500 m, and 1000 m, ending at 2300 LT September 17th, 2019 (a) and 0700 LT September 18th, 2019 (b), determined by the HYSPLIT model.**


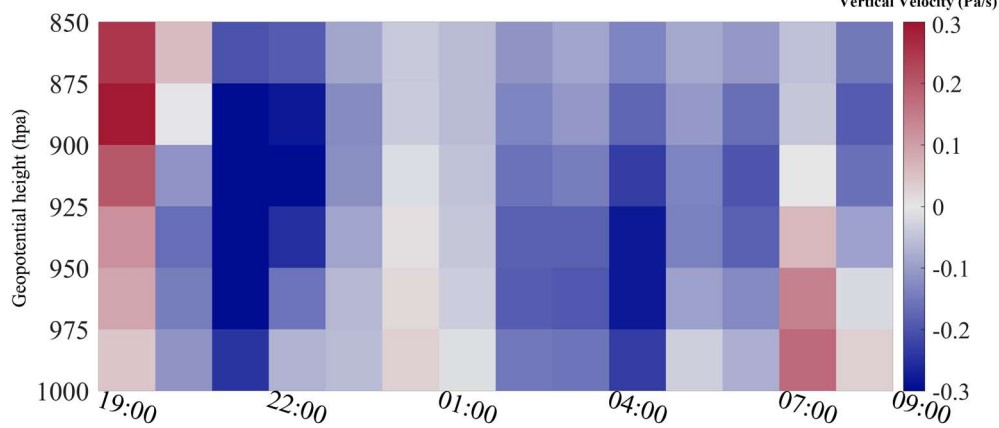

**Figure 11. ERA5 hourly vertical velocity from 1900 LT on September 17th, 2019, to 0900 LT on September**
**18th, 2019, at 23.25°N, 113.25°E. Negative values indicate upward motion.**

### 3.2.3 Type III: Double Layer

Figure 12 presents a thick single layer of particles transforming into a double layer structure. There was
a layer concentrated near the ground after 2300 LT, along with another layer suspended at the height of
600–1000 m. A cleaner layer with a lower extinction coefficient existed between the two particle
layers. The depolarization ratio of the suspending layer was higher than the layer near the surface,
especially after 0100 LT, which indicated that sources of the two layers might be different.

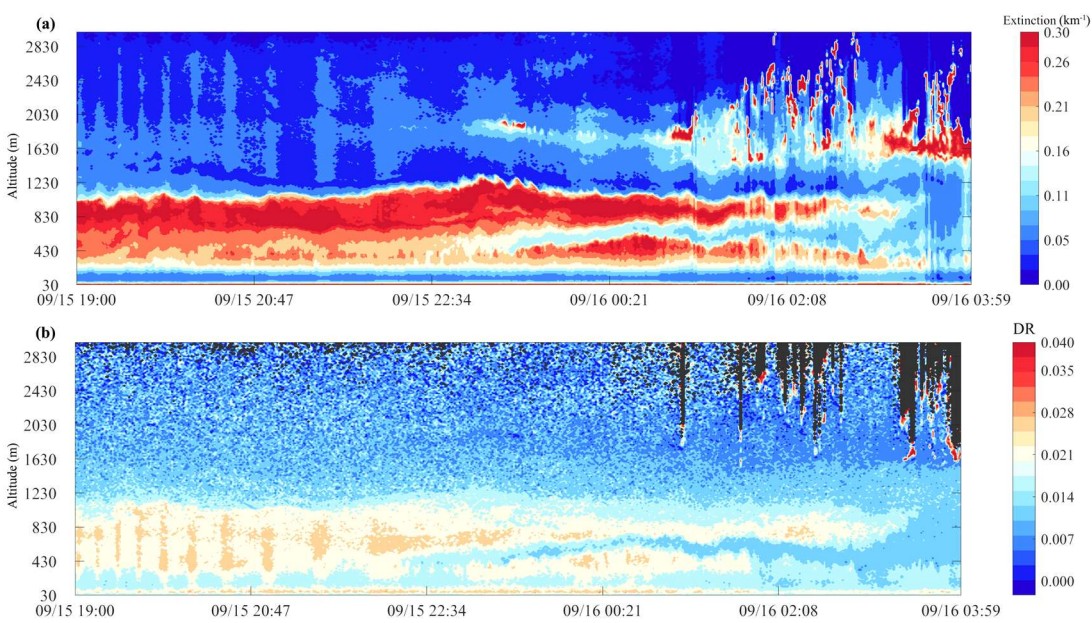

**Figure 12. Extinction coefficient at 532 nm (a) and depolarization ratio (b) from 1900 LT on September 15th,**
**2019 to 0359 LT on September 16th, 2019.**
The vertical distribution of particulate matter was closely related to the horizontal wind speed at
various heights (Figure 13). It can be seen that the wind speed of more than 1000 m increased
significantly with the altitude, reaching more than 6 m/s. By 2300 LT, the wind speed below 500 m was
approximately 4 m/s, obviously higher than the wind speed between 500–1000 m, and there were
significant differences in the wind direction. After 2300 LT, the wind speed near the ground decreased,
and wind direction gradually turned consistent with the upper level. The wind speed at 500 m
continued to be high, reaching 6 m/s maximumly. The layer with higher wind speed corresponded to
the height of the cleaner layer, which facilitated the transport of particulate matter downstream in a
horizontal direction. Figure 14 illustrates the backward trajectories when the double layer appeared. As
shown in Figure 14, the layer of particulate matter below 500 m may have originated in the southwest
of the GBA, whereas the layer of particulate matter at 1000 m may have originated from cities north of
the GBA area.

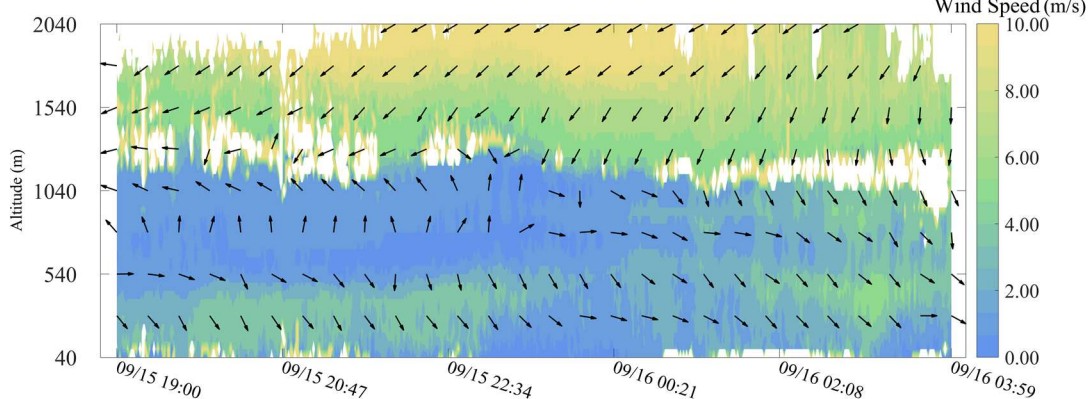

**Figure 13. Wind speed and wind direction of Type Ⅲ.**

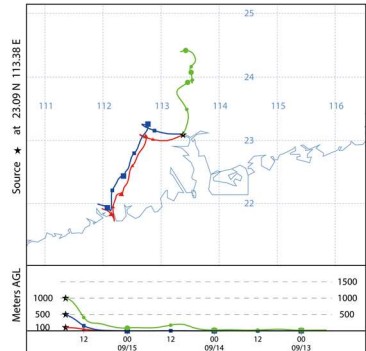

**Figure 14. Backward trajectories at 100 m, 500 m, and 1000 m, ending at 0100 LT September 16th, 2019,**
**determined by the HYSPLIT model.**
The vertical observations of the temperature (Fig. 15.) showed that on the night of September 15th,
2019, there was an inversion at 1200 m, which grew thicker. At 0048 LT, like the distribution of the
extinction coefficient, the inversion transformed into a double layer structure, with one remaining at
1200 m and another existing under 600 m. The vertical distributed double inversion, which allowed
particulate matter to concentrate at the corresponding height, resulted in a double layer distribution of
particulate matter.

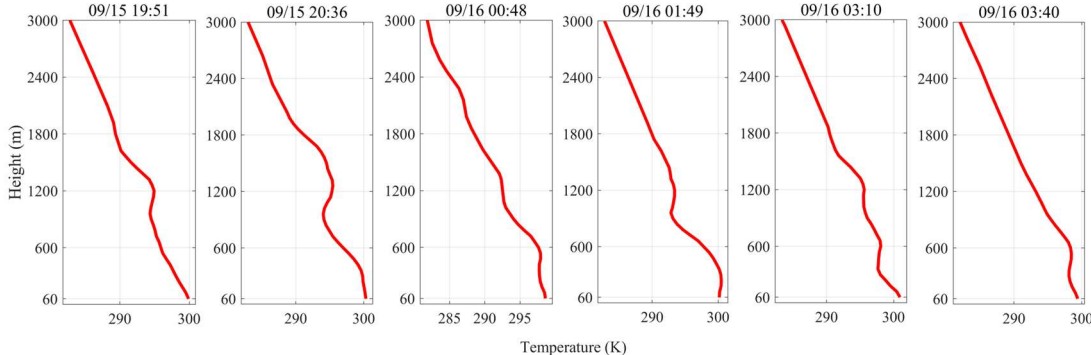

**Figure 15. Temperature profiles from the evening of September 15th, 2019 to the early hours of September**
**16th, 2019.**

**3.3    Effect of Meteorological Elements on the Distribution of Particles**
**3.3.1    Extinction Coefficient at Different Wind Speeds**
Using data of in situ observations during September and October of 2019 and 2020, statistics of
average extinction coefficients at different altitudes and horizontal wind speeds were gathered, as
shown in Figure 16. To eliminate the influence of clouds on the extinction coefficient, observations
during cloudy weather were manually screened out based on the original signal of the MPL output and
images of the sky above the field taken automatically by a camera. Because the spatial resolutions of
the data from the two lidar are different, we interpolated the data to make them match each other
vertically. The result shows that 500 m was the height with the highest average extinction coefficient,
which indicated that the particle layer was most likely to appear at this height. The horizontal wind
speed had different effects on the lower and upper parts of the boundary layer. Below 800 m, the
extinction coefficient decreased as the wind speed increased, but it was the opposite above 800 m; i.e.,
the extinction coefficient increased with the wind speed. This altering of the extinction coefficient was
because most of the particulate matter in the lower layer came from local emissions and easily
accumulated in the presence of a layer with calm wind near the ground. However, in the upper layer,
particulate matter was derived more from the surrounding areas, necessitating a certain minimum
horizontal wind speed before it could be transported by the wind.

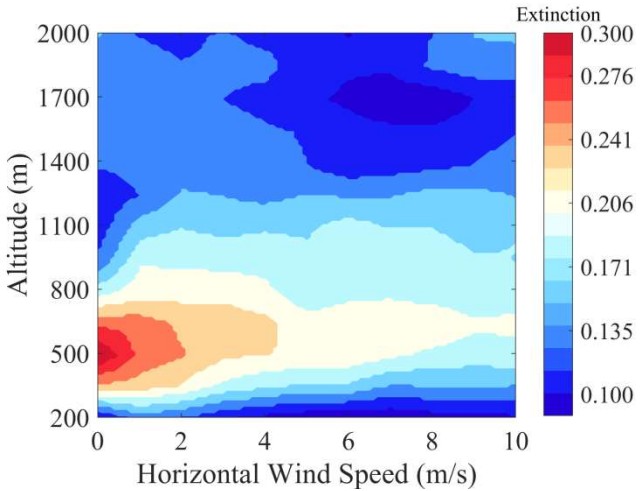


**Figure 16. Average extinction coefficient at different wind speeds and altitude from fixed-location observations of a total of 89 days at Haizhu Lake Research Base.**

### 3.3.2 Conceptual Model of Meteorological Elements and Vertical Distribution of Particles

Based on the observational research above, a conceptual model was developed to summarise the effect of meteorological elements on the three typical vertical distributions of particles in the GBA.

As shown in Figure 17, the surface single layer occurred when light horizontal wind dominated near the ground, which was not conducive to removing particles from local emissions. An elevated single layer was caused by upward airflow near the ground and vertical wind shear at a higher altitude. In this kind of wind structure, particle layer formation was dominated by upward convection and regional transport. A double layer existed because a layer with stronger horizontal wind existed between two layers with weaker wind, which facilitated the transport of particles from local emission and horizontal transport to downstream areas and resulted in a cleaner layer inside the polluted air mass.

Another key factor that influenced the vertical distribution of particles was temperature inversion, which trapped most anthropogenic emissions from the surface, preventing them from penetrating out of the boundary layer. Furthermore, multiple inversions can cause more than one peak in the concentration of particles vertically.

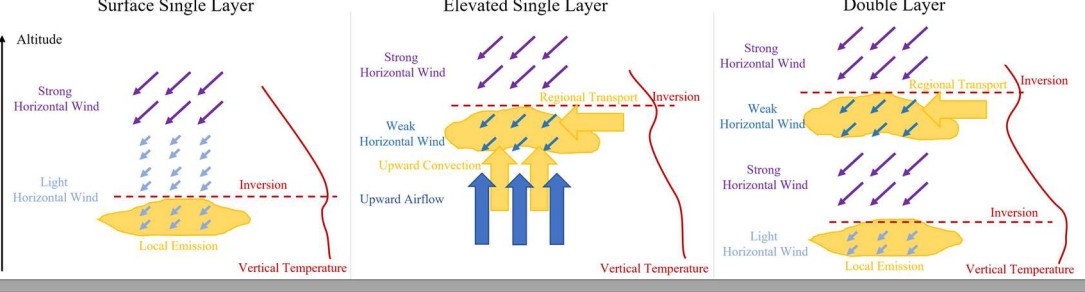

**Figure 17. Conceptual model of meteorological elements and vertical distribution of particles.**

### 4. Conclusion

The results of our study show how meteorological elements affected the three-dimensional distribution of particles in the western Guangdong–Hong Kong–Macao Greater Bay area. We focused mainly on the periods when the wet season changes to the dry season, as the frequently changing temperature and wind under such conditions have a more significant impact on the distribution of particles. The horizontal distribution of particles in the GBA was closely related to wind speed and wind direction. On days with stronger winds in the boundary layer, high values of AOD were mostly distributed in the downstream areas. On days with weaker winds, the horizontal distribution of particles in the GBA was homogeneous. The vertical distribution of particles in the GBA was classified into three typical types: surface single layer, elevated single layer, and double layer. The surface single layer occurred when

wind with very low speed dominated the boundary layer. The elevated single layer was caused by upward airflow near the ground and vertical wind shear at a higher altitude. The double layer existed because a layer with higher horizontal wind speed existed between two layers with weaker wind. Particles were concentrated near the temperature inversion. Multiple inversions can cause more than one peak in the vertical distribution of particulate matter. The mechanisms that affected the distribution of particulate matter in the upper and lower boundary layers are different. Lower horizontal wind speed was conducive to accumulating particulate matter near the ground, whereas higher horizontal wind speed promoted the transport of particles between surrounding areas in the upper boundary layer.

Further studies should be conducted during other seasons in the western Guangdong–Hong Kong–Macao Greater Bay Area to further verify the conceptual model of meteorological elements and vertical distribution of particles proposed in this article. In addition, more vertical observation instruments for meteorological elements, such as a radiometer, could be added to the multi-lidar system to further study the influence of the three-dimensional distribution of humidity, air pressure, and other meteorological elements on the distribution of particles.

**Acknowledgments**

This work was supported by the National Natural Science Foundation of China (Grant No. 41630422) and the Guangdong Major Project of Basic and Applied Basic Research (Grant No. 2020B0301030004).

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
