# Peer review of "Measurement report: Vehicle-based Multi-lidar"

_Atmospheric Chemistry and Physics, 2021_

## Author Comment (AC1)

Reply to Anonymous Referee # 1 Comment on 'Measurement report: Vehicle-based and In Situ Multi-lidar Observational Study on the Effect of Meteorological Elements on the Three-dimensional Distribution of Particles in the Western Guangdong–Hong Kong–Macao Greater Bay Area'

**comments:**

The measurement report 'Vehicle-based and In Situ Multi-lidar Observational Study on the Effect of Meteorological Elements on the Three-dimensional Distribution of Particles in the Western Guangdong–Hong Kong–Macao Greater Bay Area' focuses on the analysis of data collected by different lidar systems which were mounted on a vehicle during several day-time research tours in the western Guangdong–Hong Kong–Macao Greater Bay Area in China. This dataset is complemented by ground based lidar measurements during night-time at the Haizhu Lake research base. The goal of the study is to characterize the typical vertical distributions of aerosols over the research area.

I consider such a measurement campaign as an important contribution to the understanding of pollution in Chinese metropolitan regions. However, the determination of three different types of vertical aerosol distributions from just three night-time lidar measurements in one specific month of the year seems to be cherry-picking and inadequate. For a possible publication in ACP I wished for a longer measurement period and an in-depth analysis of the observations which may fortify the stated hypotheses in this manuscript. In my opinion, the current manuscript is more like a data-paper or paper of first observations which may be published elsewhere in more appropriate journals like for example ESSD.

In addition to that I found much room for improvement when reading through the manuscript, i.e. in the description of the employed methods, in the presentation of the analyzed data, the interpretation of the collected data as well as in the discussion of the results. Unfortunately, some parts of the manuscript were also hard to review because of poor English language.

I strongly encourage the authors to thoroughly revise the submitted manuscript with regard to these points for a possible resubmission. I also encourage the authors to involve a native speaker for proofreading before resubmitting.

This is why I tried to do my best in outlining the specific points that motivated my decision and that should be considered when revising the paper.

**Reply:**

We would like to thank you for the careful review of our manuscript. We have thoroughly revised the manuscript based on all your comments and suggestions. We sincerely appreciate your comments on language errors. We are currently revising the manuscript and checked carefully to avoid typos.

One of your main points was that the determination of three different types of vertical aerosol distributions from three night-time lidar measurements in one specific month of the year seems to be inadequate. We feel sorry that our description of the experiment was not sufficient enough, which caused your misunderstanding. We conducted vehicle-based lidar observation experiments from September 10 to October 8, 2019, and from August 29 to October 27, 2020. During 89 days of continuous observation, and we found that most of the aerosol vertical distributions are consistent with these three distribution types. Due to space limitations, we only selected the three most representative processes for analysis. Details about the measurement has also been added to the manuscript.

**General comments:**

**comments:**

As lidar is an active remote sensing technique and not an in-situ measurement technique, the phrase 'in-situ' in the manuscript/title is misleading.

**Reply:**

We have replaced 'in-situ' with 'fixed-location'. At the same time, we apologize for the misleading. All lidar data used in this manuscript are derived from vehicle-based lidars, including data of fixed-location observations and mobile observations.

**comments:**

The abstract should be written in present tense

**Reply:**

We have rewritten the abstract into the present tense.

**comments:**

The instrument platform has to be better described. In the current form of the manuscript the reader has no idea of the setup and the vehicle that was used. What type of car did you use? How have the instruments been mounted onto the car? How fast did the vehicle go? I guess you did not drive the vehicle with constant speed, but you had to adapt the speed of the vehicle to the traffic condition.

**Reply:**

Thank you for your suggestion! We have added information about the instrument platform. The car we used was a modified 7-seater Mercedes-Benz sport utility vehicle. Three lidars were fixed to the rear of the car by steel bars to ensure their stability. In order to avoid the impact of frequent changes in speed and vehicle bumps on the observation, the routes were basically flat highways, and the driving speed was controlled within 80 km/h.

**comments:**

The measurements of the research trips are only described on a half-page. I wished for an in-depth description and discussion of the collected data. What was the motivation for the chosen route that

you drove along? How long does it take to drive the 320 km? Why did you only go during daytime? When did you start with the measurement-circle? In the morning hours? I can imagine that it makes a difference if you start measuring in the morning hours compared to starting in the afternoon hours.

**Reply:**

Thank you for your suggestion! We have added more information about our measurements. The reason for choosing this route is that it covers the major urban agglomerations in the western part of the Guangdong-Hong Kong-Macao Greater Bay Area, which contains a large number of aerosol anthropogenic emission sources. It is representative for studying the regional distribution of particles in this area. We conducted mobile observations once a day, from August 29 to September 4, 2020. The set off time was at 10:00 and a single measurement-circle was completed at around 16:00. Due to the surface heating, convection in the boundary layer develops vigorously during daytime, which allows the aerosol to mix well in the boundary layer and form a more homogeneous distribution vertically. Therefore, it is more suitable to conduct mobile observations during the daytime to study the horizontal distribution of particles in the GBA area. Meanwhile, according to our observations, aerosols are more likely to have different types of vertical distribution nocturnally, which motivated us to continuously conduct fixed-location observations at night on days that we made a mobile observation in the daytime. Details of our observations were shown in the table below.

| Time                                    | Observation                |
|-----------------------------------------|----------------------------|
| Sept. 10 – Oct. 8, 2019                 | Fixed-location observation |
| Aug. 29 – Sept. 4, 2020, in the daytime | Mobile observation         |
| Aug. 29 – Sept. 4, 2020, at night       | Fixed-location observation |
| Sept. 5 – Oct. 27, 2020                 | Fixed-location observation |

**comments:**

Even though you added the links to the webpages of the respective lidar systems, I wished for a more detailed introduction of the used instruments as well as of the instrument setup on the vehicle. What are the measurement uncertainties? What are the limitations of the instruments? How did the setup on the vehicle look like?

**Reply:**

We are currently collecting relevant information from manufacturers. We have previously compared simultaneous observations from two meteorological observation towers in Beijing and Shenzhen with the lidar system. The results show that observations of this lidar system was highly accurate and suitable for studying. We have also used this lidar system in our previous research and have shown them to be reliable (He et al., 2021a; He et al., 2021b). The setup on the vehicle was shown in the figure below.

He, Y., Xu, X., Gu, Z., Chen, X., Li, Y., and Fan, S.: Vertical distribution characteristics of aerosol particles over the Guanzhong Plain, Atmospheric Environment, 255, 118444, https://doi.org/10.1016/j.atmosenv.2021.118444, 2021a.

He, Y., Wang, H., Wang, H., Xu, X., Li, Y., & Fan, S.: Meteorology and topographic influences on nocturnal ozone increase during the summertime over Shaoguan, China, Atmospheric Environment, 256, 118459, https://doi.org/10.1016/j.atmosenv.2021.118459, 2021b.

**comments:**

Have the ground-based lidar observations at Haizhu Lake Research Base been conducted with the same instruments that were also mounted onto the vehicle? Why don't you show time-height lidar plots from data collected on the vehicle? Why was some of the data collected in September 2019 and some of the data collected in September 2020? What was the motivation to choose only the month of September?

**Reply:**

We have added more information about the equipment. All our fixed-location observations and mobile observations were done using the same vehicle-based lidar system. During fixed-location observations, the car was parked in the observation field and connected to a stable power source. As we pay more attention to the horizontal distribution of aerosols during the mobile observations, and also limited by space, we did not show time-height lidar plots. We conducted lidar observation experiments from September 10 to October 8, 2019, and from August 29 to October 27, 2020. The reason for choosing these two months is that September and October are the periods when the wet season changes to the dry season in the GBA area. Therefore, changes in meteorological elements have a significant impact on the three-dimensional distribution of particles.

**comments:**

Although this paper is a measurement report, it's vague to define three types of aerosol distributions from single observations in autumn. Would long-term lidar measurements at the Haizhu Lake Research Base be available to conduct a statistical analysis of the typical aerosol layering over different seasons of the year in the region? This would substantiate your hypotheses.

**Reply:**

Thank you for your suggestion! The Haizhu Lake Research Base is located in the centre of the metropolis Guangzhou, so it is representative for studying the distribution of urban aerosols. Unfortunately, there is only ground observation equipment in the base. This motivated us to park the car in the base and conducted a total of 89 days of fixed-location observation. During 89 days of continuous observation, we found that most of the aerosol vertical distributions are consistent with these three distribution types. Due to space limitations, we only selected the three most representative processes for analyzing the three distribution types. Statistical analysis was conducted when studying the extinction coefficient at different wind speeds in 3.3.1, using the fix-location observation data of 89 days.

**comments:**

Could you also discuss the possible impact of the topography on your measurements? The research area seems to be a basin surrounded by a quite hilly/mountainous region.

**Reply:**

That is a very good suggestion. The research area is on the Pearl River Delta Plain. The area is bordered by the Nanling Mountains in the north. The obstruction of mountains makes the GBA area less susceptible to long-distance transport of pollutants from other areas. The transport of pollutants mainly occurs between cities in the research area. We have added this content to the manuscript.

**comments:**

You could go into more detail with analyzing the lidar data. What aerosol types do you observe? Could it be possible that marine sea salt contributes to the observed aerosol layers as you have observed a southerly component of the wind speed at low altitudes (especially during Type 1).

**Reply:**

Thank you for your suggestion! We used micro pulse lidar for aerosols observation. The obtained extinction coefficient generally reflects the strength of light absorption by the total aerosol in the air. It cannot distinguish the type of different aerosols. In this study, we are more concerned about the anthropogenic particulate matter in the metropolis, therefore sea salt aerosols are not in our focus. Fortunately, the CALIPSO satellite passed just a few hours before Type 1 over the observation area, so we also plotted the corresponding VFM aerosol subtype data from the CALIPSO satellite inversion, and the results show that there is a low level of sea salt aerosols in our region of interest.

However, in future studies we will also consider adding equipment for measuring aerosol components.

Nearly all figures and their captions have to be revised, as many things remain unclear to the reader (for details see below).

Specific comments:

**comments:**

Figure 1: For a better comparison and to condense the information shown you could use wind barbs and plot them in Figure 4.

**Reply:**

Thank you for your suggestion. We couldn't agree more. We are currently revising the images based on this suggestion.

**comments:**

Formula 1: Please motivate why you have chosen a fixed value of S = 50 sr for the conversion to the extinction coefficient.

**Reply:**

S = 50 sr was the default value given by the manufacturer. This value was consistent with prior work in the GBA area (Li et al., 2020).

Li, Y., Wang, B., Lee, S. Y., Zhang, Z., Wang, Y., and Dong, W.: Micro-Pulse Lidar Cruising Measurements in Northern South China Sea, Remote Sensing, 12(10), 1695, https://doi.org/10.3390/rs12101695, 2020.

**comments:**

What is k representing in formula (2). What depolarization ratio are you exactly measuring? According to the formula it is the volume linear depolarization ratio.

**Reply:**

Thank you for your suggestion! k is the depolarization calibration constant, which is the ratio of the gains of the parallel and perpendicular channels (Dai et al., 2018). This content has been added to the manuscript.

Dai, G., Wu, S., and Song, X.: Depolarization ratio profiles calibration and observations of aerosol and cloud in the Tibetan Plateau based on polarization Raman lidar, Remote Sensing, 10(3), 378, https://doi.org/10.3390/rs10030378, 2018.

**comments:**

Line 192: '...wind direction over the observation points...' I don't understand what points you mean. Which instrument was used for the measurements? Please clarify.

**Reply:**

This should be 'location' rather than 'points'. We are sorry that we have made a mistake in our wording and have caused you to misunderstand. The instrument used for this measurement is the Doppler wind profile lidar.

**comments:**

Line 195: How exactly can low wind speeds at low altitudes act as a disincentive for regional transport at higher altitudes? Please explain in more detail.

**Reply:**

We would like to express that low wind speeds near the surface favour the accumulation of locally generated particles and that low wind speeds at higher altitudes (500-1000 m) are not conducive to the transport of particles from surrounding areas over the observation location. We have revised the manuscript to avoid confusion.

**comments:**

Line 208: Why did you choose 22 LT as starting time for your trajectory calculations? From the shown wind measurements, you already see that later in the night the wind (at 540 m) shifts to the South. I could imagine that a starting point later in the night would show a completely different result.

**Reply:**

We have calculated every time of the process. The results of HYSPLIT calculation are shown in the figure below. HYSPLIT model calculations based on reanalysis data yielded that the trajectory came from the north at all times of the process. Due to space constraints, we have selected only one representative figure. This also shows that the use of remote sensing instruments such as Doppler wind lidar could capture changes that are not included in the reanalysis data.

---

## Author Comment (AC2)

Reply to Anonymous Referee # 2 Comment on 'Measurement report: Vehicle-based and In Situ Multi-lidar Observational Study on the Effect of Meteorological Elements on the Three-dimensional Distribution of Particles in the Western Guangdong–Hong Kong–Macao Greater Bay Area'

General comments:

This manuscript attempts to understand the mechanism of how wind and temperature in the boundary layer affects the horizontal and vertical distribution of particles. The topic is critical, and the method is scientifically sound. I suggest accepting the publication after the following revisions.

We would like to thank the Anonymous Referee # 2 for the assessment of our manuscript and for sound and constructive comments. We took into all account comments and suggestions, and performed revision of the manuscript, trying to clarify all issues. The referee's comments are in black; our responses are in dark blue.

Major comments:

Line 54-60: The introduction part could be improved by providing discussion about the temperature and wind impacts on aerosol distribution, citing relevant work that used multiple lidars to study temperature, wind and aerosol, and discussing their findings.

Thank you for your suggestion! We have improved the relevant content. This part has been modified as follows (changes to the manuscript are indicated in red font):

"The distribution of particles is influenced not only by changes in source emissions but also by changes in meteorological factors, such as temperature and wind. It has previously been observed that a low boundary layer height and complex vertical distributions of aerosols, temperature and relative humidity were the main structural characteristics of haze days (Huige et al., 2021). Previous studies have confirmed that different types of temperature inversions have different impacts on particles in the boundary layer (Wallace et al., 2009; Wang et al., 2018). The depth and temperature difference of the inversion is a key factor in the predictions of surface $PM_{2.5}$ concentrations (Zang et al., 2017). It has previously been observed that wind fields play an important role in transboundary-local aerosols interaction (Huang et al., 2021a; Huang et al., 2021b). Recent evidence suggests that wind shear was an important factor in terms of $PM_{10}$ vertical profile modification (Sekuła, P., et al). The concentration of particulate matter also shows characteristics of wind-dependent spatial distributions in which pollutant transport within the GBA city cluster is significant (Xie et al., 2019). Hence, the issue of how meteorological factors affecting the distribution of particles has received considerable critical attention."

Line 99-111: it would be useful if the authors can describe how to keep the different spatial and time resolutions of the three kinds of lidar systems consistent in this study.

Since the spatial resolutions of the data from the micro pulse lidar and the Doppler wind profile lidar are different, we interpolated the data to make them match each other vertically when

calculating extinction coefficient at different wind speeds in 3.3.1. As for the data from the Raman temperature profile lidar, we maintain the original spatial and temporal resolution when plotting. This information has been added to 3.3.1.

A brief discussion about the uncertainties would be useful.

Thank you for your suggestion! The quality of data from the lidar system was checked before using in our study. Results show that the percentage difference between data provided by the lidar system and data from the Shenzhen meteorological tower was less than 15%, which indicates a sufficient accuracy of the lidar instrument. This information has been added to 2.2.

Line 170: "Therefore, the value of the extinction coefficient near the ground during the day was generally low…". This is an interesting statement as the surface layer PM2.5 concentrations during daytime are typically higher than nighttime (average). The aforementioned statement seems to give a different perspective. It would be great if the authors can explain this a bit.

According to our observations before, the concentration of particulate matter near the ground is generally lower during the day than at night. Boundary layer height is generally higher in the daytime. The high solar radiation intensity during the day results in significant surface warming and thus stronger thermal convection in the vertical direction. Therefore, daytime conditions are more favorable for the vertical dispersion of pollutants. However, concentrations of particulate matter are also related to local emission intensities and regional transport, and observations may vary from region to region.

Minor comments:

Line 73: insufficient reference to support line 72: in the past few years, several …

We have added more reference of this topic.

Line 121: what is the value of the Zc in this study?

Zc is variable, ranging from 10-15 km, and depending on the signal intensity. This information has been added to 2.3.

Line 131-136: what is the horizontal resolution of the meteorology data used in HYSPLIT?

We used the meteorological data of the Global Data Assimilation System (GDAS) at the spatial resolution of 0.25° in HYSPLIT. This information has been added to 2.4.

Line 184: It would be useful to describe how to observe the layer of elevated depolarization ratio layer in Fig 3(a)?

We are sorry that we made a mistake. In this version of the manuscript it should be Figure 4(b). we

have modified it. Thank you very much for the correction!

"Wind speed at lower altitudes was relatively low, which was beneficial to regional transport…" should be further elaborated.

Thank you for your suggestion! We have modified this sentence as follows (changes to the manuscript are indicated in red font):
The domination of weak wind in the boundary layer was beneficial to inter-city transport of particles. It brought particles from cities located upstream, to the location of our observation, and allowed particles to stay longer without being blown quickly downstream.

---

## Author Response (AR2)

Reply to Anonymous Referee # 1 Comment on 'Measurement report: Vehicle-based and In Situ Multi-lidar Observational Study on the Effect of Meteorological Elements on the Three-dimensional Distribution of Particles in the Western Guangdong–Hong Kong–Macao Greater Bay Area'

The measurement report 'Vehicle-based and In Situ Multi-lidar Observational Study on the Effect of Meteorological Elements on the Three-dimensional Distribution of Particles in the Western Guangdong–Hong Kong–Macao Greater Bay Area' focuses on the analysis of data collected by different lidar systems which were mounted on a vehicle during several day-time research tours in the western Guangdong–Hong Kong–Macao Greater Bay Area in China. This dataset is complemented by ground based lidar measurements during night-time at the Haizhu Lake research base. The goal of the study is to characterize the typical vertical distributions of aerosols over the research area.

I consider such a measurement campaign as an important contribution to the understanding of pollution in Chinese metropolitan regions. However, the determination of three different types of vertical aerosol distributions from just three night-time lidar measurements in one specific month of the year seems to be cherry-picking and inadequate. For a possible publication in ACP I wished for a longer measurement period and an in-depth analysis of the observations which may fortify the stated hypotheses in this manuscript. In my opinion, the current manuscript is more like a data-paper or paper of first observations which may be published elsewhere in more appropriate journals like for example ESSD.

In addition to that I found much room for improvement when reading through the manuscript, i.e. in the description of the employed methods, in the presentation of the analyzed data, the interpretation of the collected data as well as in the discussion of the results. Unfortunately, some parts of the manuscript were also hard to review because of poor English language.

I strongly encourage the authors to thoroughly revise the submitted manuscript with regard to these points for a possible resubmission. I also encourage the authors to involve a native speaker for proofreading before resubmitting.

This is why I tried to do my best in outlining the specific points that motivated my decision and that should be considered when revising the paper.

We would like to thank you for the careful review of our manuscript. We have thoroughly revised the manuscript based on all your comments and suggestions. We sincerely appreciate your comments on language errors. We have revised the manuscript and checked carefully to avoid typos.

One of your main points was that the determination of three different types of vertical aerosol distributions from three night-time lidar measurements in one specific month of the year seems to be inadequate. We feel sorry that our description of the experiment was not sufficient enough, which caused your misunderstanding. We conducted vehicle-based lidar observation experiments from September 10 to October 8, 2019, and from August 29 to October 27, 2020. During 89 days of continuous observation, and we found that most of the aerosol vertical distributions are consistent

with these three distribution types. Therefore, we selected the three most representative processes for analysis. Details about the measurement has also been added to the manuscript in 2.1.

General comments:

As lidar is an active remote sensing technique and not an in-situ measurement technique, the phrase 'in-situ' in the manuscript/title is misleading.

We have replaced all the phrase 'in-situ' with 'fixed-location'. At the same time, we apologize for the misleading. All lidar data used in this manuscript are derived from vehicle-based lidars, including data of fixed-location observations and mobile observations. To avoid misleading, the title has been changed to "Measurement report: Vehicle-based Multi-lidar Observational Study of the Effect of Meteorological Elements on the Three-dimensional Distribution of Particles in the Western Guangdong–Hong Kong–Macao Greater Bay Area".

The abstract should be written in present tense

We have rewritten the abstract into correct tense.

The instrument platform has to be better described. In the current form of the manuscript the reader has no idea of the setup and the vehicle that was used. What type of car did you use? How have the instruments been mounted onto the car? How fast did the vehicle go? I guess you did not drive the vehicle with constant speed, but you had to adapt the speed of the vehicle to the traffic condition.

Thank you for your suggestion! We have added information about the instrument platform. This part has been modified as follows (information added to the manuscript are indicated in red font):

"A multi-lidar system was installed on a vehicle in this experiment. The car used was a modified 7-seater Mercedes-Benz sport utility vehicle. Three lidars were fixed to the rear of the car by steel bars to ensure their stability. To avoid the impact of frequent changes in speed and vehicle bumps during the observation, the routes of mobile observations were basically flat highways, and the driving speed was controlled within 80 km/h. During fixed-location observations, the car was parked in the observation field and connected to a stable power source."

The measurements of the research trips are only described on a half-page. I wished for an in-depth description and discussion of the collected data. What was the motivation for the chosen route that you drove along? How long does it take to drive the 320 km? Why did you only go during daytime? When did you start with the measurement-circle? In the morning hours? I can imagine that it makes a difference if you start measuring in the morning hours compared to starting in the afternoon hours.

Thank you for your suggestion! We have added more information about the mobile measurements in 3.1 as follows (information added to the manuscript are indicated in red font):

"The horizontal distribution of particles was obtained by conducting mobile vehicle lidar observations

in the GBA. The reason for choosing this route is that it covers the major urban agglomerations in the western part of the Guangdong–Hong Kong–Macao Greater Bay Area, which contains a large number of anthropogenic aerosol emission sources. It is representative of the regional distribution of particles in this area. We conducted mobile observations once a day, from August 29th to September 4th, 2020. The set off time was at 10:00 and a single measurement circle was completed at around 16:00. Owing to surface heating, convection in the boundary layer develops vigorously during daytime, which allows aerosols to mix well and form a more homogeneous vertical distribution. Therefore, mobile observations during the daytime are more appropriate to study the horizontal distribution of particles in the GBA area."

To clearly show the time of the experiment, we have added a table in 2.1 as follow:

| Time | Observation |
|---|---|
| Sept. 10 – Oct. 8, 2019 | Fixed-location observation |
| Aug. 29 – Sept. 4, 2020, in the daytime | Mobile observation |
| Aug. 29 – Sept. 4, 2020, at night | Fixed-location observation |
| Sept. 5 – Oct. 27, 2020 | Fixed-location observation |

Even though you added the links to the webpages of the respective lidar systems, I wished for a more detailed introduction of the used instruments as well as of the instrument setup on the vehicle. What are the measurement uncertainties? What are the limitations of the instruments? How did the setup on the vehicle look like?

The instruments have been compared with meteorological towers prior to the observation tests. We have added a discussion about the limitations and uncertainties in 2.2. This part has been modified as follows (changes to the manuscript are indicated in red font):

"The lidar system included a 3D visual scanning micro pulse lidar (EV-Lidar-CAM, EVERISE Company, Beijing, http://www.everisetech.com.cn/products/ygtc/evlidarportable.html), a twirling Raman temperature profile lidar (TRL20, EVERISE Company, Beijing, http://www.everisetech.com.cn/products/ygtc/templidar.html), a Doppler wind profile lidar (Windview10, EVERISE Company, Beijing, http://www.everisetech.com.cn/products/ygtc/windview10.html), a global positioning system (GPS), and a signal acquisition unit. The three lidars are characterised by high temporal and spatial resolution and can effectively determine the evolution of the vertical distribution of particles, as well as temperature, wind speed, and wind direction over time. Remote sensing observations, such as lidar, are generally considered to be less accurate than direct observations from aircraft and meteorological tower-based sensors. Therefore, the quality of data from the lidar system was checked before using in our study. Results show that the percentage difference between data provided by the lidar system and data from the Shenzhen meteorological tower was less than 15%, which indicates a sufficient accuracy of the lidar instrument. We have used this lidar system in our previous research and showed it to be reliable (He et al., 2021a; He et al., 2021b). The vehicle setup is shown in Figure 2. Details of the three lidars are shown in Table 2."

He, Y., Xu, X., Gu, Z., Chen, X., Li, Y., and Fan, S.: Vertical distribution characteristics of aerosol

particles over the Guanzhong Plain, Atmospheric Environment, 255, 118444, https://doi.org/10.1016/j.atmosenv.2021.118444, 2021a.

He, Y., Wang, H., Wang, H., Xu, X., Li, Y., & Fan, S.: Meteorology and topographic influences on nocturnal ozone increase during the summertime over Shaoguan, China, Atmospheric Environment, 256, 118459, https://doi.org/10.1016/j.atmosenv.2021.118459, 2021b.

We have added a figure showing the setup on the vehicle in 2.2 as follows:

[Figure]

**Figure 2. Setup of the multi-lidar system on the vehicle.**

Have the ground-based lidar observations at Haizhu Lake Research Base been conducted with the same instruments that were also mounted onto the vehicle? Why don't you show time-height lidar plots from data collected on the vehicle? Why was some of the data collected in September 2019 and some of the data collected in September 2020? What was the motivation to choose only the month of September?

We have added more information about the equipment. All our fixed-location observations and mobile observations were done using the same vehicle-based lidar system. During fixed-location observations, the car was parked in the observation field and connected to a stable power source. As we pay more attention to the horizontal distribution of aerosols during the mobile observations, and also limited by space, we did not show time-height lidar plots. We conducted lidar observation experiments from September 10 to October 8, 2019, and from August 29 to October 27, 2020. The reason for choosing these two months is that September and October are the periods when the wet season changes to the dry season in the GBA area. Therefore, changes in meteorological elements have a significant impact on the three-dimensional distribution of particles. We have added our motivation of choosing this period in 2.1 in red font:

"The reason for choosing these periods is that they include the wet season change to the dry season in the GBA area. Therefore, changes in meteorological elements have a significant impact on the three-dimensional distribution of particles. The location of the Haizhu Lake Research Base and the area of the measuring path are shown in Fig. 1."

Although this paper is a measurement report, it's vague to define three types of aerosol distributions from single observations in autumn. Would long-term lidar measurements at the Haizhu Lake Research Base be available to conduct a statistical analysis of the typical aerosol layering over different seasons of the year in the region? This would substantiate your hypotheses.

Thank you for your suggestion! The Haizhu Lake Research Base is located in the centre of the metropolis Guangzhou, so it is representative for studying the distribution of urban aerosols. Unfortunately, there is only ground observation equipment in the base. This motivated us to park the car in the base and conducted a total of 89 days of fixed-location observation. During 89 days of continuous observation, we found that most of the aerosol vertical distributions are consistent with these three distribution types. Therefore, we selected the three most representative processes for analyzing the three distribution types. Statistical analysis was conducted when studying the extinction coefficient at different wind speeds in 3.3.1, using the fix-location observation data of 89 days.

Could you also discuss the possible impact of the topography on your measurements? The research area seems to be a basin surrounded by a quite hilly/mountainous region.

That is a very good suggestion. We have added discussions about impact of the topography in 2.1 in red font:

"The location of the Haizhu Lake Research Base and the area of the measuring path are shown in Fig. 1. The research area is on the Pearl River Delta Plain. This area is bordered by the Nanling Mountains in the north. Mountain obstruction makes the GBA area less susceptible to long-distance transport of pollutants from other areas, and the transport of pollutants mainly occurs between cities in the research area."

You could go into more detail with analyzing the lidar data. What aerosol types do you observe? Could it be possible that marine sea salt contributes to the observed aerosol layers as you have observed a southerly component of the wind speed at low altitudes (especially during Type 1).

Thank you for your suggestion! We used micro pulse lidar for aerosols observation. The obtained extinction coefficient generally reflects the strength of light absorption by the total aerosol in the air. It cannot distinguish the type of different aerosols. In this study, we are more concerned about the anthropogenic particulate matter in the metropolis, therefore sea salt aerosols are not in our focus. Fortunately, the CALIPSO satellite passed just a few hours before Type 1 over the observation area, so we also plotted the corresponding VFM aerosol subtype data from the CALIPSO satellite inversion, and the results show that there is a low level of sea salt aerosols in our region of interest. However, in future studies we will also consider adding equipment for measuring aerosol components.

[Figure]

Nearly all figures and their captions have to be revised, as many things remain unclear to the reader (for details see below).

Specific comments:

Figure 1: For a better comparison and to condense the information shown you could use wind barbs and plot them in Figure 4.

Thank you for your suggestion. We have tried using wind barbs mapping but found that using colours to represent wind speed gives a more visual indication of areas with small wind speeds, and the high resolution of the reanalysis data can be retained (because wind barbs cannot be drawn as densely as the colour filling). However, if required we can use the wind barbs to plot.

Formula 1: Please motivate why you have chosen a fixed value of S = 50 sr for the conversion to the extinction coefficient.

S = 50 sr was the default value given by the manufacturer. This value was consistent with prior work in the GBA area (Li et al., 2020). We have added this information in 2.3 as follow in red font:

"$S_a = 50$ Sr the particle extinction-to-backscatter ratio, which is the default value given by the manufacturer. This value is consistent with prior work in the GBA area (Li et al., 2020)."

Li, Y., Wang, B., Lee, S. Y., Zhang, Z., Wang, Y., and Dong, W.: Micro-Pulse Lidar Cruising Measurements in Northern South China Sea, Remote Sensing, 12(10), 1695, https://doi.org/10.3390/rs12101695, 2020.

What is k representing in formula (2). What depolarization ratio are you exactly measuring? According to the formula it is the volume linear depolarization ratio.

Thank you for your suggestion! k is the depolarization calibration constant, which is the ratio of the gains of the parallel and perpendicular channels (Dai et al., 2018). We have added this information in 2.3 as follow in red font:

"The depolarization ratio is calculated with the following formula:

$$\delta = k \frac{P_\perp}{P_\parallel} \qquad (2)$$

where $P_\perp$ and $P_\parallel$ represent the cross-polarized and co-polarized signal, respectively. k the depolarization calibration constant, which is the ratio of the gains of the parallel and perpendicular channels (Dai et al., 2018)."

Dai, G., Wu, S., and Song, X.: Depolarization ratio profiles calibration and observations of aerosol and cloud in the Tibetan Plateau based on polarization Raman lidar, Remote Sensing, 10(3), 378, https://doi.org/10.3390/rs10030378, 2018.

Line 192: '…wind direction over the observation points…' I don't understand what points you mean. Which instrument was used for the measurements? Please clarify.

This should be 'location' rather than 'points'. We have revised it. We are sorry that we have made a mistake in our wording and have caused you to misunderstand. The instrument used for this measurement is the Doppler wind profile lidar. Changes to the manuscript in 3.2.1 are indicated in red font:

"Figure 6 shows the horizontal wind speed and wind direction of the fixed-location observation in this period."

Line 195: How exactly can low wind speeds at low altitudes act as a disincentive for regional transport at higher altitudes? Please explain in more detail.

We would like to express that low wind speeds near the surface favour the accumulation of locally generated particles and that low wind speeds at higher altitudes (500-1000 m) are not conducive to the transport of particles from surrounding areas over the observation location. We have revised the manuscript to avoid confusion. Changes to the manuscript in 3.2.1 are indicated in red font:

"Such a static and stable condition was advantageous to the accumulation of locally generated particulate matter near the ground. However, light wind at higher altitude (500–1000 m) prevented the regional transport of particulate matter at a higher altitude, because it is difficult for such a low wind speed to blow the particulate matter at the corresponding height to the downstream area. Therefore, when light wind dominated near the ground, the particulate matter was likely to form a single layer on the surface."

Line 208: Why did you choose 22 LT as starting time for your trajectory calculations? From the shown wind measurements, you already see that later in the night the wind (at 540 m) shifts to the South. I could imagine that a starting point later in the night would show a completely different result.

We have calculated every time of the process. The results of HYSPLIT calculation are shown in the figure below. HYSPLIT model calculations based on reanalysis data yielded that the trajectory came from the north at all times of the process. Due to space constraints, we have selected only one representative figure. This also shows that the use of remote sensing instruments such as Doppler wind lidar could capture changes that are not included in the reanalysis data.

[Figure]

Line 226: Aerosol is always suspended in the air, also when it's located near surface. Please correct.

Thank you for your suggestion! We have corrected this incorrect wording. Changes to the manuscript in 3.2.2 are indicated in red font:

"The particle layer was not only distributed near the ground but sometimes suspended at a higher altitude."

Line 234: I don't understand this sentence. What 'unconverted primary particulate matter' are you meaning? Sea salt? Depolarization predominantly depends on the shape of the particles.

In this context we refer to primary pollutant emissions from anthropogenic sources near the surface, including industrial emissions, traffic emissions, etc. We have revised the manuscript to avoid confusion. According to existing studies, particle depolarization ratio from lidar is an indicator of non-spherical particles and is sensitive to the fraction of non-spherical particles and their size (Burton et al., 2015). If aerosol particles are small in size, they often show a spherical character when observing them with a micro pulse lidar, and the value of the depolarization ratio of the aerosol is small.

Changes to the manuscript in 3.2.2 are indicated in red font:

"However, the depolarization ratio of *Type II* was significantly lower than the depolarization ratio of the particle layer near the surface of *Type I*. This differing depolarization ratio was because local emissions dominated in *Type I*, and the primary pollutant emissions from anthropogenic sources near the surface with a non-spherical character and larger particle size accounted for a larger amount than that of *Type II*."

Burton, S. P., Hair, J. W., Kahnert, M., Ferrare, R. A., Hostetler, C. A., Cook, A. L., Harper, D. B., Berkoff, T. A., Seaman, S. T., Collins, J. E., Fenn, M. A., and Rogers, R. R.: Observations of the spectral dependence of linear particle depolarization ratio of aerosols using NASA Langley airborne High Spectral Resolution Lidar, Atmos. Chem. Phys., 15, 13453–13473, https://doi.org/10.5194/acp-15-13453-2015, 2015.

Chapter 'Conclusion' is rather a summary than a conclusion.

Thank you very much for your advice! We have rewritten the chapter to condense our findings.

Line 384: I guess that according to the ACP-Guidelines data should be made freely available in an online repository.

Yes, we will make the data public.

Specific comments to shown figures:

Figure 1: Y-Axis – a sequential colormap instead of a diverging colormap would fit better? Where did you take the underlying Altitude data from? It's not clear to the reader. The resolution seems to be quite coarse.

Thank you very much for your suggestion! The altitude data we used before has a resolution of 0.033 degrees. We have used data with higher resolution for the plots, as well as a sequential colormap.

[Figure]

Figure 2: Please label the colorbar and axes and add information at what wavelength AOD has been measured. Are the AOD-measurements averaged over time? Why do you only show vertically integrated point-measurements and no continuous and height-resolved measurements?

Thank you very much for your suggestion! As in this chapter we focus on the horizontal distribution of particles in the GBA area, we use AOD figure for our discussion. Due to occlusion of the laser along the route, etc., we filtered the anomalous data and calculated the averaged AOD over time for plotting. The colorbar and axes has been labeled as follow. Wavelength of the measured AOD has also been added.

[Figure]

**Figure 3. (a)-(g) Aerosol optical depth (AOD) measured at 532 nm with the MPL in the route from August 29th to September 4th, 2020, and (h) Guangdong–Hong Kong–Macao Greater Bay Area and route details.**

Figure 3: Please label the shown axes and clarify from which model the wind field has been retrieved. Could you indicate the research area in these plots? This would make it easier for the reader to compare Figure 2 to Figure 3.

Thank you very much for your suggestion! The axes have been labeled as follow. The GBA area has also been indicated with orange box. We also added the source of the ERA5 data in 3.1 in red font:

"Figure 4 shows low-level horizontal wind fields on 925 hPa over the region based on ERA5 reanalysis data (https://cds.climate.copernicus.eu/cdsapp#!/dataset/reanalysis-era5-pressure-levels?tab=overview)."

[Figure]

Figure 4. (a)-(g) Wind field of 925 hPa from August 29th to September 4th, 2020 from ERA5 reanalysis data. The colour map represents horizontal wind speed (m/s). Red arrows represent the wind direction. The orange box shows the location of the GBA area.

Figure 4: Is the figure showing lidar measurements? Or is the data collected form a model? What is the y-axis showing? Depolarization ratio at 532 nm?

This figure shows lidar measurements. Y-axis refer to the altitude above instrument. Extinction coefficient and depolarization ratio at 532 nm are shown in figure (a) and (b) respectively. To avoid misleading, we have modified the caption of this figure and labeled the Y-axis as follow:

[Figure]

Figure 5. Extinction coefficient (a) and depolarization ratio (b) at 532 nm from 2154 LT on September 2nd, 2020, to 0609 LT on September 3rd, 2020.

Figure 5: Clarify meaning of Y-axis? Altitude above sea level or above instrument? Why does it only show 4 h of data and not 8 h like the lidar data in Figure 4?

Y-axis refers to the altitude above instrument. Due to equipment failure, the Doppler wind profile lidar was not able to capture data after 03:00, so we only show data before 02:33. We have labeled the Y-axis as follow:

[Figure]

Figure 6. Wind speed and wind direction of Type I. Colour map represents horizontal wind speed (m/s). Arrows represents the wind direction.

We also added information of the altitude in 3.2 to avoid misleading in red font:

"Three different vertical distribution types of particles are given below, as well as the corresponding vertical observation results of temperature and wind in the same period. Altitude values in the following figures refer to the altitude above instrument."

Figure 7: Please use SI-units: Kelvin instead of degrees Celsius and clarify meaning of Y-axis. Altitude above sea level or above vehicle? Please indicate the measurement uncertainty of the Raman-Temperature measurements.

Thank you very much for your suggestion! Discussion of the uncertainty has been added in 2.2. Y-axis refer to the altitude above instrument. We have changed the unit into SI-units in the manuscripts. The figure has been revised as follow:

[Figure]

**Figure 8. Temperature profiles from the evening of September 2nd, 2020 to the early hours of September 3rd, 2020.**

Figure 8: Same suggestions as for Figure 4.

This figure also shows lidar measurements. Y-axis refer to the altitude above instrument. Extinction coefficient and depolarization ratio at 532 nm are shown in figure (a) and (b) respectively. We have modified this figure as follow:

[Figure]

**Figure 9. Extinction coefficient (a) and depolarization ratio (b) at 532 nm from 1900 LT on September 17th, 2019 to 0859 LT on September 18th, 2019.**

Figure 10: What is going on between 19 pm LT and 19:30 pm LT? Please modify the range of the colorbar to resolve the magnitude of the apparent updraft.

Thanks to your correction, we have adjusted the plotting method and the result is shown below. We can see that the area was controlled by downdrafts until 20:00, after 20:00 it changes to updrafts.

[Figure]

**Figure 11. ERA5 hourly vertical velocity from 1900 LT on September 17th, 2019, to 0900 LT on September 18th, 2019, at 23.25°N, 113.25°E. Negative values indicate upward motion.**

Figure 11: Same suggestions as for Figure 4. Where are these periodic oscillations in the first half of the night in the lidar signals coming from? Is this a natural phenomenon or a measurement-artefact?

This figure also shows lidar measurements. Y-axis refer to the altitude above instrument. We consider these periodic oscillations as a measurement-artefact rather than a natural phenomenon.

We have modified this figure as follow:

[Figure]

**Figure 12. Extinction coefficient (a) and depolarization ratio (b) at 532 nm from 1900 LT on September 15th, 2019 to 0359 LT on September 16th, 2019.**

Figure 12: Same suggestions as for Figure 5.

Y-axis refer to the altitude above instrument. We have modified this figure as follow:

[Figure]

**Figure 13. Wind speed and wind direction of *Type III*.**

Figure 14: Same suggestions as for Figure 7.

Thank you very much for your suggestion! Y-axis also refer to the altitude above instrument. We have changed the unit into SI-units. The figure has been revised as follow:

[Figure]

**Figure 15. Temperature profiles from the evening of September 15th, 2019 to the early hours of September 16th, 2019.**

Figure 15: From where has the wind speed data been taken? Please clarify in caption.

Thank you for your suggestion! Wind speed data was obtained from fixed-location observations at Haizhu Lake Research Base using the Doppler wind profile lidar. We have added the information in caption as follow:

[Figure]

**Figure 16. Average extinction coefficient at different wind speeds and altitude from fixed-location observations of a total of 89 days at Haizhu Lake Research Base.**

Reply to Anonymous Referee # 2 Comment on 'Measurement report: Vehicle-based and In Situ Multi-lidar Observational Study on the Effect of Meteorological Elements on the Three-dimensional Distribution of Particles in the Western Guangdong–Hong Kong–Macao Greater Bay Area'

General comments:

This manuscript attempts to understand the mechanism of how wind and temperature in the boundary layer affects the horizontal and vertical distribution of particles. The topic is critical, and the method is scientifically sound. I suggest accepting the publication after the following revisions.

We would like to thank the Anonymous Referee # 2 for the assessment of our manuscript and for sound and constructive comments. We took into all account comments and suggestions, and performed revision of the manuscript, trying to clarify all issues. The referee's comments are in black; our responses are in dark blue.

Major comments:

Line 54-60: The introduction part could be improved by providing discussion about the temperature and wind impacts on aerosol distribution, citing relevant work that used multiple lidars to study temperature, wind and aerosol, and discussing their findings.

Thank you for your suggestion! We have improved the relevant content. This part has been modified as follows (changes to the manuscript are indicated in red font):

"The distribution of particles is influenced not only by changes in source emissions but also by changes in meteorological factors, such as temperature and wind. It has previously been observed that a low boundary layer height and complex vertical distributions of aerosols, temperature and relative humidity were the main structural characteristics of haze days (Huige et al., 2021). Previous studies have confirmed that different types of temperature inversions have different impacts on particles in the boundary layer (Wallace et al., 2009; Wang et al., 2018). The depth and temperature difference of the inversion is a key factor in the predictions of surface $PM_{2.5}$ concentrations (Zang et al., 2017). It has previously been observed that wind fields play an important role in transboundary-local aerosols interaction (Huang et al., 2021a; Huang et al., 2021b). Recent evidence suggests that wind shear was an important factor in terms of $PM_{10}$ vertical profile modification (Sekuła, P., et al). The concentration of particulate matter also shows characteristics of wind-dependent spatial distributions in which pollutant transport within the GBA city cluster is significant (Xie et al., 2019). Hence, the issue of how meteorological factors affecting the distribution of particles has received considerable critical attention."

Line 99-111: it would be useful if the authors can describe how to keep the different spatial and time resolutions of the three kinds of lidar systems consistent in this study.

Since the spatial resolutions of the data from the micro pulse lidar and the Doppler wind profile lidar are different, we interpolated the data to make them match each other vertically when

calculating extinction coefficient at different wind speeds in 3.3.1. As for the data from the Raman temperature profile lidar, we maintain the original spatial and temporal resolution when plotting. This information has been added to 3.3.1.

A brief discussion about the uncertainties would be useful.

Thank you for your suggestion! The quality of data from the lidar system was checked before using in our study. Results show that the percentage difference between data provided by the lidar system and data from the Shenzhen meteorological tower was less than 15%, which indicates a sufficient accuracy of the lidar instrument. This information has been added to 2.2.

Line 170: "Therefore, the value of the extinction coefficient near the ground during the day was generally low…". This is an interesting statement as the surface layer PM2.5 concentrations during daytime are typically higher than nighttime (average). The aforementioned statement seems to give a different perspective. It would be great if the authors can explain this a bit.

According to our observations before, the concentration of particulate matter near the ground is generally lower during the day than at night. Boundary layer height is generally higher in the daytime. The high solar radiation intensity during the day results in significant surface warming and thus stronger thermal convection in the vertical direction. Therefore, daytime conditions are more favorable for the vertical dispersion of pollutants. However, concentrations of particulate matter are also related to local emission intensities and regional transport, and observations may vary from region to region.

Minor comments:

Line 73: insufficient reference to support line 72: in the past few years, several …

We have added more reference of this topic.

Line 121: what is the value of the Zc in this study?

Zc is variable, ranging from 10-15 km, and depending on the signal intensity. This information has been added to 2.3.

Line 131-136: what is the horizontal resolution of the meteorology data used in HYSPLIT?

We used the meteorological data of the Global Data Assimilation System (GDAS) at the spatial resolution of 0.25° in HYSPLIT. This information has been added to 2.4.

Line 184: It would be useful to describe how to observe the layer of elevated depolarization ratio layer in Fig 3(a)?

We are sorry that we made a mistake. In this version of the manuscript it should be Figure 4(b). we

have modified it. Thank you very much for the correction!

"Wind speed at lower altitudes was relatively low, which was beneficial to regional transport…" should be further elaborated.

Thank you for your suggestion! We have modified this sentence as follows (changes to the manuscript are indicated in red font):
The domination of weak wind in the boundary layer was beneficial to inter-city transport of particles. It brought particles from cities located upstream, to the location of our observation, and allowed particles to stay longer without being blown quickly downstream.